# Viral entry shapes HCMV latency establishment

Yaarit Kitsberg ®[1], Aharon Nachshon ®[1], Tamar Arazi[1], Karin Broennimann[1], Tal Fisher[1], Alexander Wainstein[1,2], Yaara Finkel ®[1,3], Noam Stern-Ginossar ®[1] ✉ & Michal Schwartz ®[1] ✉

Human cytomegalovirus (HCMV) infection results in either productive or latent infection, the latter enabling life-long viral persistence. Monocytes support latent infection but become permissive to productive infection upon differentiation into macrophages. These differentiation-driven differences have been largely attributed to chromatin-mediated repression of the viral genome. Using metabolic labeling of newly synthesized RNA, we observe markedly lower viral transcription at early stages of infection in monocytes compared to macrophages. Unbiased comparison reveals that this difference is partly explained by inefficient viral entry in monocytes: fewer viruses enter, and correspondingly, fewer genomes reach the nucleus. Indeed, ectopic expression of known HCMV entry receptors in monocytes enhances viral entry and enables productive infection, demonstrating that these cells can support full lytic replication if entry is efficient. We further identify integrin β3 as a differentiation-induced surface protein playing an important role in HCMV entry into macrophages, partially accounting for the observed differences in entry efficiency. Finally, we show that cells receiving fewer viral genomes are the ones that establish latent infection and have the capacity to reactivate. Overall, our findings reveal that entry is a previously unrecognized factor contributing to latent infection in monocytes, adding a critical layer to the paradigm of HCMV latency.

Human Cytomegalovirus (HCMV) is a human beta-herpesvirus infecting the majority of the population worldwide. Like other herpesviruses, HCMV persists through the lifetime of its host by establishing latency. Cells of the hematopoietic system were identified as key sites for HCMV latency. CD34+ hematopoietic stem cells (HSCs), early progenitors of the myeloid system in the bone marrow, and blood monocytes are the main cell types in which HCMV latency has been characterized[1–3]. In contrast, terminally differentiated myeloid cells, such as macrophages and dendritic cells are considered permissive for productive HCMV infection, and differentiation of infected monocytes to these cell types can lead to viral reactivation[4–7].

scRNA-seq works in differentiated myeloid cells have shown that initial high expression of the viral immediate early genes, namely IE1 and IE2, reflects progression towards productive infection[8–10]. The accepted underlying assumption is that chromatin-dependent repression of the viral genome, and specifically of immediate early genes, is the basis for latent infection in monocytes and HSCs[11], and that differentiation leads to distinct chromatin deposition that enables viral gene expression[12,13]. However, the molecular roots for these differences in repression upon differentiation are not well understood.

Here, using metabolic labeling of newly synthesized mRNA, we reveal that compared to macrophages, early viral gene expression in

[1]Department of Molecular Genetics, Weizmann Institute of Science, Rehovot, Israel. [2]Present address: Department of Molecular Cell Biology, Weizmann Institute of Science, Rehovot, Israel. [3]Present address: Department of Bioengineering, Stanford University School of Medicine, Stanford, CA, USA. ✉e-mail: noam.stern-ginossar@weizmann.ac.il; michalsc@weizmann.ac.il

undifferentiated monocytic cells is much lower. By systematically comparing monocytes and their differentiated counterparts, we show that this strong difference in gene expression is partially due to differences in viral entry and that in monocytes, fewer viruses enter the cells and correspondingly, fewer viral genomes reach the nucleus. Remarkably, ectopic expression of known HCMV entry receptors in monocytes results in more efficient viral entry and concomitantly, productive infection. We further show that integrin β3, which was previously associated with HCMV entry[14], is upregulated upon differentiation and plays a critical role in macrophage infection. Importantly, we demonstrate that infected cells that receive fewer viral genomes are the source of latent cells that can later reactivate. Overall, we uncover that inefficient entry, due to specific cell surface composition, is a major factor precluding productive infection in monocytes, and that this inefficient entry leads to cells receiving low levels of viral genomes leading to the establishment of latent infection.

## Results

### The early phase of HCMV infection in monocytes features strikingly low viral gene transcription

Monocytes are known to be latently infected with HCMV and do not support productive infection, however, following differentiation they become permissive to productive infection[4,5]. Indeed, we could recapitulate these differentiation-based differences in HCMV infection in primary monocytes isolated from peripheral blood as well as in the myeloid cell lines THP1 and Kasumi-3, which are commonly used as cell models to study HCMV latency (Fig. 1a). We infected these cell types with an HCMV TB40-E strain containing a GFP reporter (HCMV-GFP), which allows convenient quantification of productively infected cells[15]. In the monocytic cells, GFP expression remained low (Figs. 1b and S1a), and correspondingly, no infectious progeny production was detected at 10 days post-infection (dpi) (Fig. 1c). Differentiation of infected monocytes at 7 dpi led to reactivation of the virus in a portion of the cells (Fig. S1b), indicating that they are indeed latently infected. Differentiation of primary, THP1 or Kasumi-3 monocytes to monocyte-derived macrophages prior to infection, resulted in a distinct population of cells that expressed high levels of GFP at 3 dpi, indicating productive infection (Figs. 1b and S1a) and indeed these cells produced infectious progeny (Fig. 1c).

We recently showed that initial levels of viral gene expression are a major factor dictating productive infection in macrophages[10], and therefore wanted to assess whether the strong differences in infection between monocytes and macrophages are reflected in substantial differences in the levels of initial viral gene transcription. Since our work and that of others demonstrated that during early infection, newly transcribed RNA is masked by virion-associated input RNA[16,17], we aimed to directly measure early viral gene transcription. We applied thiol (SH)-linked alkylation for metabolic sequencing of RNA (SLAM-seq)[18], which facilitates the measurement of newly transcribed RNA based on 4-thiouridine (4sU) incorporation into newly synthesized RNA. After RNA is extracted, 4sU is converted to a cytosine analog using iodoacetamide, and these U-to-C conversions are identified and quantified by RNA sequencing. We applied SLAM-seq to both primary and THP1 monocytes and macrophages, starting labeling at 3 h post-infection (hpi) for 2 and 3 h (cells were harvested at 5 and 6 hpi). We confirmed that 4sU labeling does not affect cell viability (Fig. S1c) and successfully generated SLAM-seq libraries in both primary and THP1-derived cells, with over 5389 genes quantified and a high U-to-C conversion rate (Fig. S1d). The levels of newly synthesized cellular transcripts were comparable between infected primary monocytes and macrophages, as well as between infected THP1 monocytes and THP1 macrophages, indicating there are no major biases in our labeling (Fig. S1e). Remarkably, newly synthesized viral transcripts were extremely low in monocytes regardless of the labeling time, while in macrophages new viral transcripts were detected after 2 h of labeling and

their relative fraction further increased at 3 h of labeling (Fig. 1d). These newly synthesized viral transcripts in macrophages were predominantly immediate early genes (UL122 and UL123), reflecting the initiation of an infection cycle (Fig. S1f). These results demonstrate that already at very early stages of infection in monocytes (both primary and THP1), viral genes are weakly transcribed, while in macrophages they are efficiently expressed.

### Cell surface proteins are upregulated upon monocyte to macrophage differentiation

The immediate vast difference in the levels of synthesized viral transcripts in infected monocytes, compared to macrophages, indicates a major difference in HCMV's ability to initiate gene expression in these two cell types. To unbiasedly search for candidate factors that may explain these dramatic differences, we performed RNA-seq on primary monocytes, THP1 monocytes and Kasumi-3 myeloid progenitor cells, as well as on macrophages derived from the same cells. Thousands of genes were differentially expressed upon differentiation (Supplementary Data 1). Pathway enrichment analysis in each of the cell types (primary monocytes, THP1 and Kasumi-3 cells) revealed that common pathways change upon differentiation of THP1 and Kasumi-3 cells, while different pathways change upon differentiation of primary monocytes (Fig. 2a). In primary monocytes, the most significantly differential pathways were related to inflammation and innate immunity that mainly decreased upon differentiation, with interferon response pathways being the most significantly reduced (Fig. 2a). This is in line with our previous work showing that intrinsic expression of interferon stimulated genes (ISGs) is decreased upon differentiation, and that this reduction contributes to the increased susceptibility to infection of macrophages compared to monocytes[10]. In THP1 and Kasumi-3, which are tumor-derived cell lines, the most significant changes were decrease in pathways related to cell proliferation, such as E2F targets, Myc targets and G2/M checkpoint (Fig. 2a), consistent with previous reports of significant reduction in these cells' proliferation capacity upon differentiation[19,20]. The effect of cell proliferation on HCMV infection has been extensively characterized in fibroblast infections[21,22]. These findings suggest that in THP1 and Kasumi-3, differences in permissivity following differentiation may be partially attributed to reduction in proliferative capacity. Significantly enriched pathways that were shared between all three cell types included a reduction in apical junction and in myogenesis, both pathways not intuitively related to myeloid differentiation processes or to early viral gene expression.

Parsimoniously, we expect the same mechanism to explain the difference in infection upon differentiation in all three cell types. Thus, we focused on common differentially expressed genes across cell types. Upon differentiation, 213 genes were commonly upregulated between primary, THP1 and Kasumi-3 cells, which is a significant overlap ($p < 10^{-5}$, Fig. S2a), and 42 genes were commonly downregulated in all three cell types ($p = 0.133$, Fig. S2a). Pathway enrichment analysis on the commonly upregulated genes yielded, as expected, several pathways related to differentiation and maturation of immune cells and immune signaling (Fig. S2b). Since we revealed massive differences in initial viral gene expression, we focused on processes that can potentially explain these differences. Although there is a major focus in the field on chromatin-related factors that regulate HCMV repression in monocytes[23,24], such factors, as a group, were not significantly enriched in the shared genes (Fig. S2c). Nevertheless, four chromatin-related factors were downregulated upon differentiation in the three cell types, including CHD3, which is implicated in the repression of the HCMV genome through the recruitment of HDACs[25,26]. We therefore explored the potential involvement of histone deacetylation, which is reported to play a key role in the repression of viral genes during HCMV latency[27,28]. We tested the ability of the potent HDAC inhibitor, TrichostatinA (TSA), which is

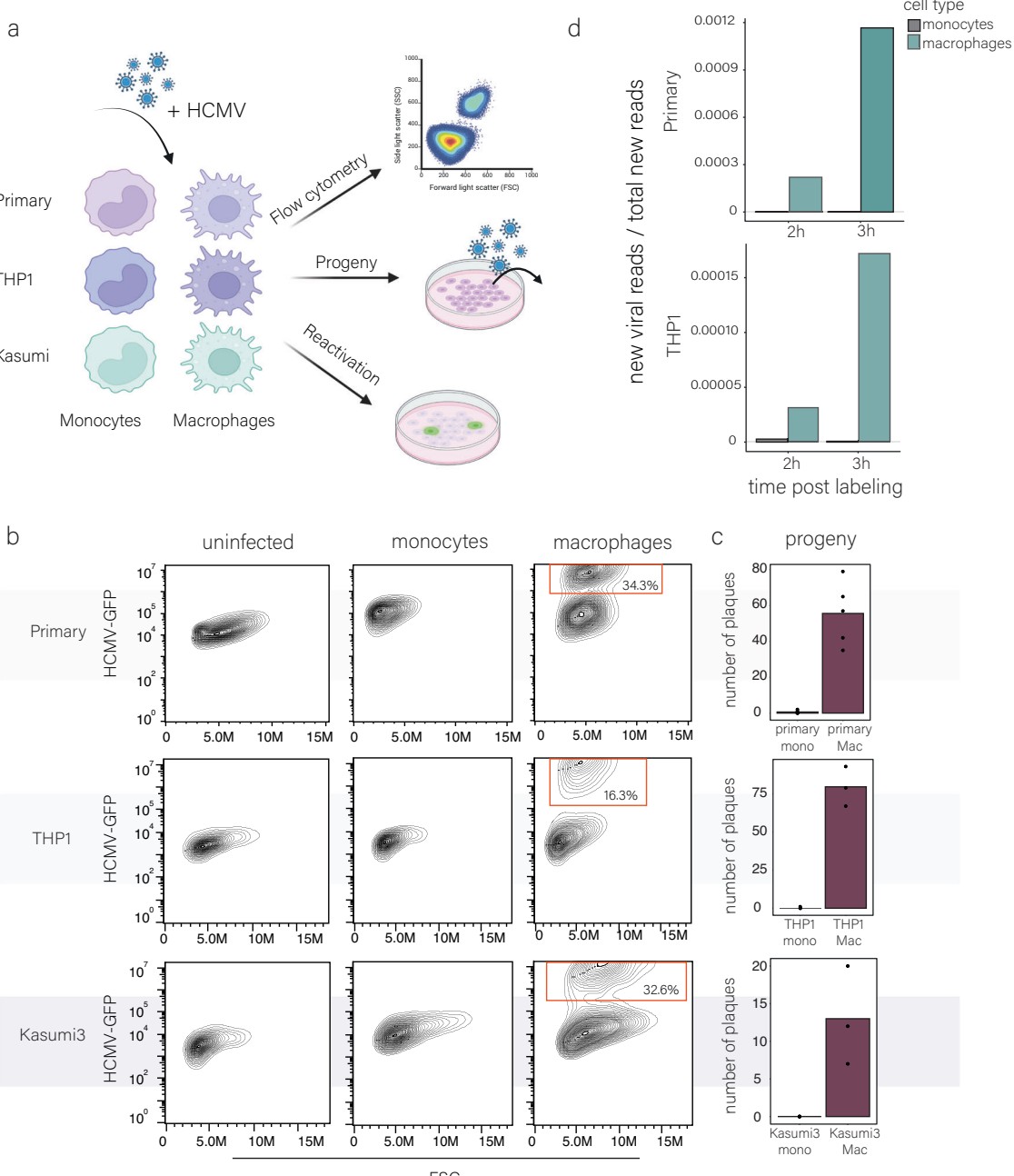

**Fig. 1 | HCMV infection and viral gene expression in monocytes and macrophages. a** Schematic illustration of the experimental setup. Monocytes and macrophages from primary, THP1 and Kasumi-3 cells were infected with HCMV-GFP. Infection levels, viral progeny, and reactivation were then analyzed. Created in BioRender. Schwartz, M. (2025) https://BioRender.com/mfrmzq3. **b** Flow cytometry analysis of primary, THP1 and Kasumi-3 monocytes and their differentiated counterparts infected with HCMV-GFP. Analysis was performed at 3 days post-infection (dpi). The red gate marks the productive, GFP-bright cell population. FSC forward scatter, M million. **c** Measurements of infectious virus in supernatants collected from infected monocytes and macrophages at 10 dpi. Mono monocytes; Mac macrophages. n = 5 in primary cells and n = 3 in THP1 and Kasumi3 cells. **d** Proportion of new viral reads out of the total new reads detected by SLAM-seq in infected primary and THP1 monocytes and macrophages. Infected cells were labeled with 4sU at 3 hpi. and harvested for SLAM-seq after 2 h (left bars) or 3 h (adjacent right bars) of labeling. Source data are provided as a Source data file.

known to induce expression of IE (immediate early) genes in THP1 cells[29], to induce productive infection in monocytes. While TSA treatment indeed led to an increase in the percentage of productively infected monocytes, a comparable effect was also observed in macrophages. Furthermore, the TSA effect was small compared to the effect of differentiation, suggesting that additional factors likely contribute to the differences between monocytes and macrophages (Figs. 2b and S2d).

Intriguingly, we observed a significant upregulation of cell surface proteins, following monocyte-to-macrophage differentiation (p = 0.042; Fig. 2c), suggesting substantial remodeling of the cell surface composition. Given that such changes can influence viral entry and, consequently, viral gene expression, we asked whether surface proteins previously implicated in HCMV entry are among the upregulated genes. Indeed, HCMV-associated entry factors were significantly enriched among the commonly upregulated surface proteins

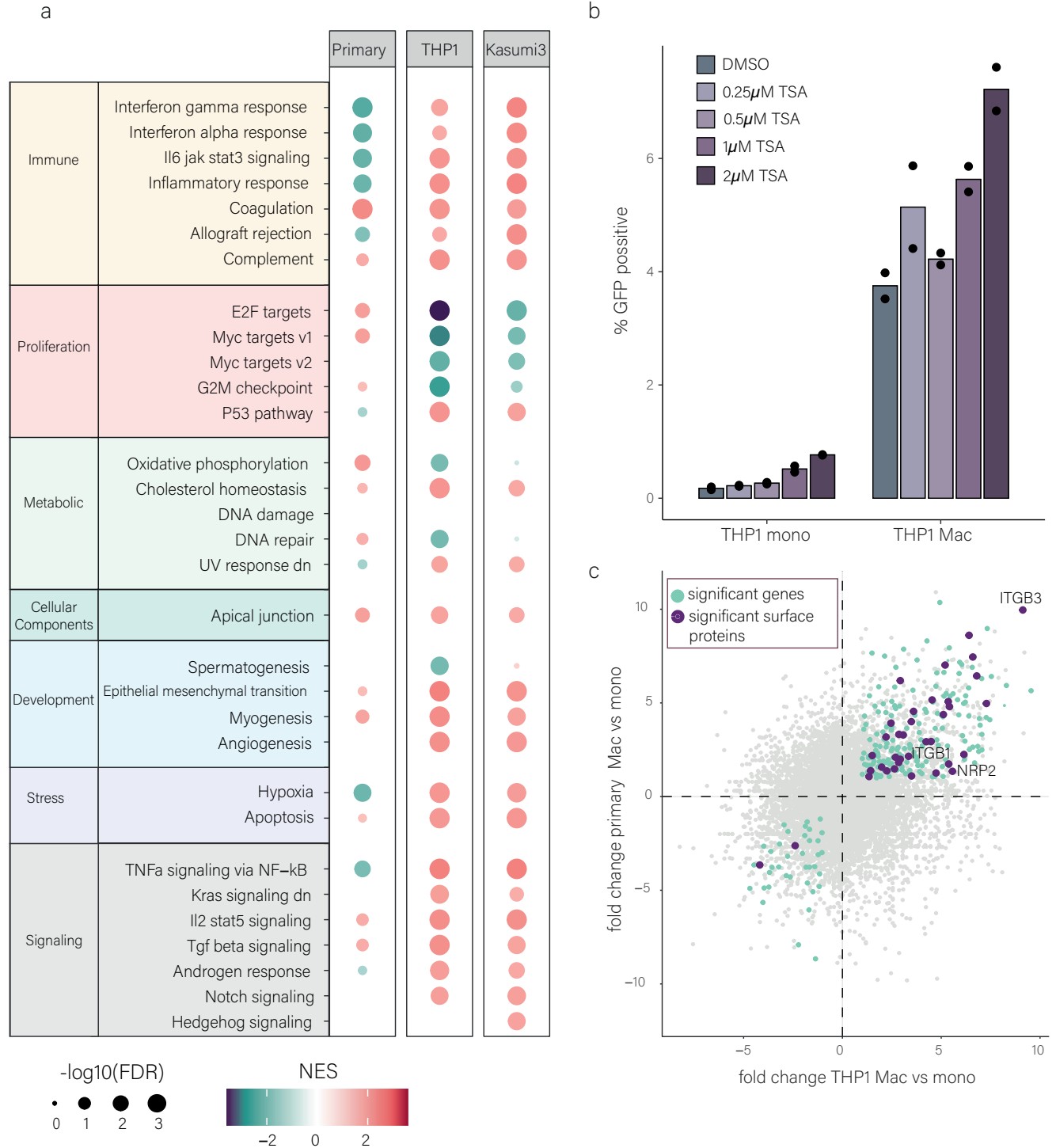

**Fig. 2 | Changes in gene expression upon differentiation. a** Summary of Hallmark pathway enrichment analysis of differentially expressed genes upon monocyte to macrophage differentiation in primary, THP1 and Kasumi-3 cells. FDR false discovery rate; NES normalized enrichment score. **b** THP1 monocytes and macrophages were treated with increasing concentrations of TSA. Cells were treated with TSA or DMSO control at 5 hpi and analyzed at 3 dpi by flow cytometry. n = 2. Gating strategy is shown in Fig. S1a. Source data are provided as a Source data file. **c** Scatterplot of the fold change (FC) from RNA-seq data between primary monocytes and macrophages, relative to the fold change of THP1 monocytes and macrophages. Light blue dots mark significantly changing genes (FDR < 0.05, LFC > 1) in all three cell types. Dark purple dots mark single transmembrane genes that are significantly changing in all three cell types (P = 0.042). Names of significantly changing cell surface proteins in all three cell types involved in HCMV entry, according to a list compiled based on[31] and other studies (see Supplementary Data 2) are shown.

(p = 0.0062, hypergeometric enrichment test; Supplementary Data 2 and Fig. 2c). These upregulated genes include NRP2, which mediates HCMV entry into non-fibroblasts cells (through the viral pentamer entry complex)[30] as well as two integrins, ITGB1 and ITGB3, which were shown to play a role in HCMV entry[31] (Fig. 2c and Supplementary Data 2). These changes in cell surface protein involved in HCMV entry, pointed to possible unexplored differences in viral entry between monocytes and macrophages.

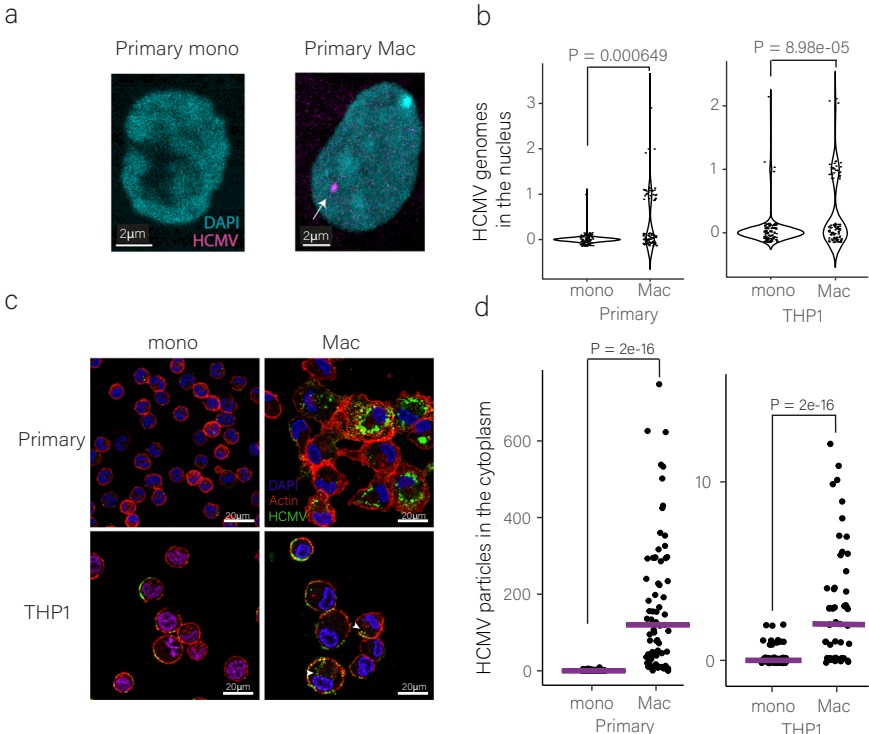

**Fig. 3 | HCMV genomes are detected at very low levels in the nucleus and cytoplasm of infected monocytes compared to macrophages. a** Images of infected primary monocyte and macrophage nuclei at 12 hpi. The HCMV genome was probed using DNA-FISH. **b** Quantification of viral genomes detected in the nuclei of infected primary or THP1 monocytes (n = 87 and n = 112, respectively) and macrophages (n = 93 and n = 109, respectively) by DNA-FISH at 12 hpi. The P-value was calculated using Poisson regression. Source data are provided as a Source data file. **c** Images of HCMV particles (labeled by UL32-GFP) in infected primary and THP1 monocytes and macrophages at 1 hpi. Actin staining was used to visualize cell borders, and DAPI for the nuclei. **d** Quantification of viral particles within the cytoplasm of infected primary and THP1 monocytes (n = 115 and n = 78, respectively) and macrophages (n = 71 and n = 53, respectively) at 1 hpi (presented in **c**). Viral particles were counted using FIJI image processing, and statistical analysis was performed using Poisson regression. Mono monocytes; mac macrophages. Source data are provided as a Source data file.

## Inefficient viral entry into monocytes is a major cause for low viral gene expression

To explore if indeed disparities in viral entry may explain some of the difference in viral gene expression, we quantified the number of viral genomes in the nuclei of infected primary and THP1 monocytes and macrophages by DNA FISH. At 12 hpi, we could detect very low counts of viral genomes in the nuclei of monocytes, much lower than the amount of viral genomes in the nuclei of macrophages (Figs. 3a; S3, b and Supplementary Movie 1). To further examine differences in entry efficiency, we utilized a virus in which the tegument protein UL32 is tagged with GFP (UL32-GFP[32]), allowing fluorescent tracking of viral particles. In agreement with our DNA-FISH measurements, we found significantly more viral particles within infected macrophages compared to infected monocytes (Fig. 3c, d). These results show there is a considerable difference in the efficiency of viral entry between monocytes and macrophages, with fewer viral genomes reaching the nucleus of monocytes.

To test whether inefficient viral entry contributes to non-productive infection of monocytes, we aimed to increase viral entry efficiency and test the effect on infection. To this end, we ectopically expressed PDGFRα, a well-characterized entry receptor of HCMV in fibroblasts[33], which is not expressed in either monocytes or macrophages (Fig. S4a), in THP1 monocytes (THP1-PDGFRα, Fig. S4b, c). Remarkably, infection of THP1-PDGFRα with HCMV-GFP resulted in a distinct population of GFP-bright cells at 3 dpi, indicating productive infection (Fig. 4a). Furthermore, productively infected cells (marked by bright GFP expression) were those with higher surface expression levels of PDGFRα suggesting a direct connection between entry and the ability to establish productive infection (Fig. 4b). Infected THP1-

PDGFRα supported viral genome replication (Fig. 4c) and generated viral replication compartments (Fig. S4d). However, viral titers were extremely low (Fig. S4e), and as shown below this is likely due to the constitutively expressed PDGFRα interfering with the infectivity of viral progeny.

To substantiate that PDGFRα expression in THP1 monocytes enhances viral entry, we quantified viral genomes at 12 hpi and found that in contrast to the parental THP1, in THP1-PDGFRα viral genomes reach the nucleus in a considerable portion of the cells (Fig. 4d and Supplementary Movie 1). Furthermore, infected THP1-PDGFRα monocytes had significantly more viral particles in the cytoplasm than THP1 monocytes (Fig. 4e), resembling the levels observed in THP1-derived macrophages (Fig. S4f). Correspondingly, these cells transcribe viral genes at 5 hpi, as measured by SLAM-seq (Fig. 4f). Notably, differentiation of THP1-PDGFRα cells further enhanced viral entry, suggesting that, as expected, differentiation facilitates HCMV entry through a PDGFRα-independent pathway (Fig. S4f). Importantly, overexpression of PDGFRα did not result in differentiation of the cells, as the cells did not differ from the parental THP1 cells morphologically or in their expression levels of macrophage-related surface markers (Fig. S4g, h). We further performed transcriptomic as well as pro-teomic analyses on THP1-PDGFRα and the parental cells and found only minor changes in gene expression and protein composition, none of which are related to macrophage differentiation or innate immunity (Fig. S4i, j and Supplementary Data 3), negating the possibility of indirect effects of PDGFRα expression.

To further rule out indirect effects of long-term PDGFRα expression and to analyze if constitutive PDGFRα expression interferes with the generation of viral progeny, we repeated the experiment with a

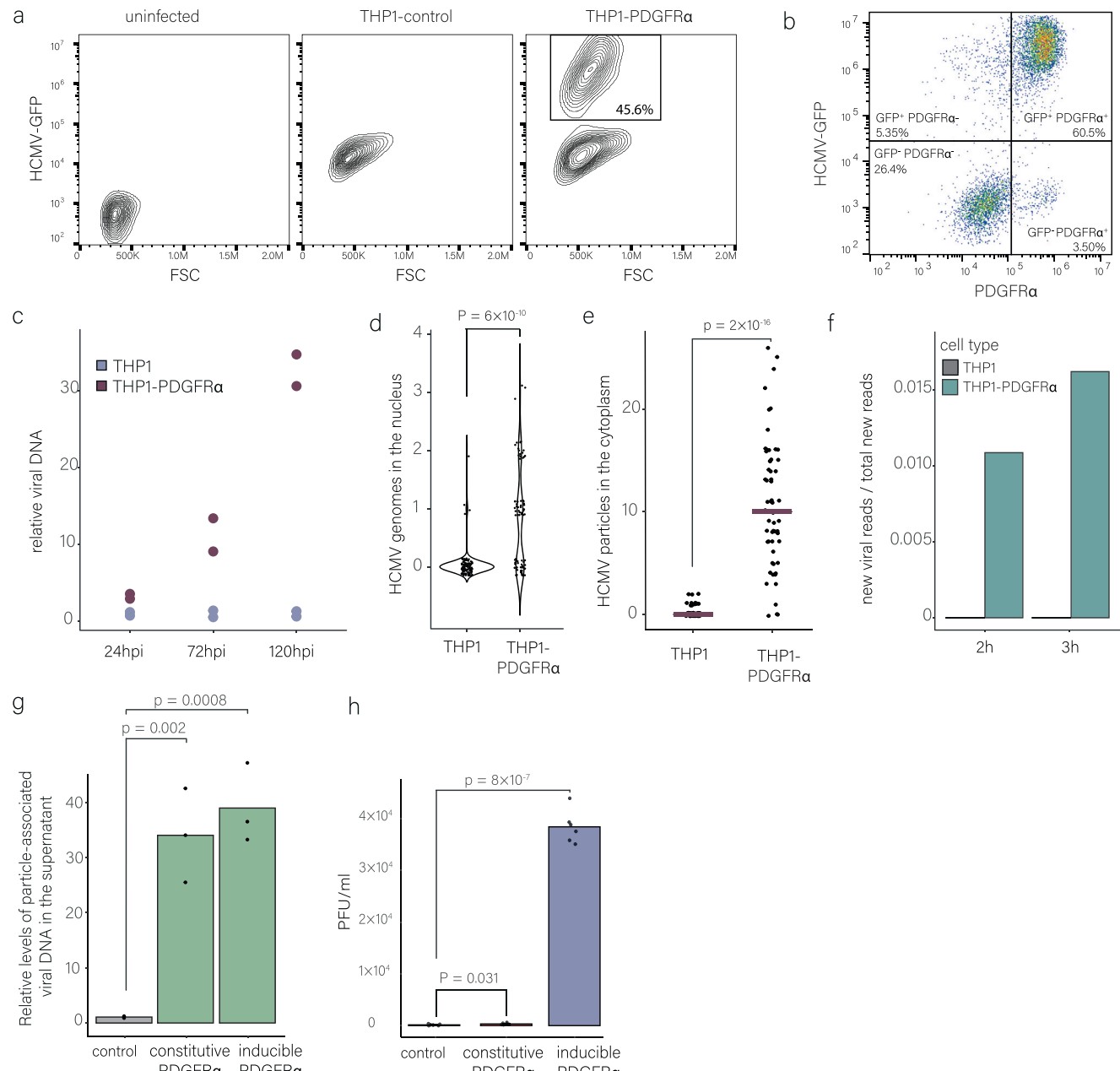

**Fig. 4 | Ectopic expression of known HCMV entry receptors in monocytes leads to productive infection. a** Flow cytometry analysis of THP1 and THP1-PDGFRα infected with HCMV-GFP at 3 dpi. **b** Flow cytometry analysis of THP1-PDGFRα infected with HCMV−GFP at 3 dpi showing HCMV-GFP level versus PDGFRα surface level. **c** Viral DNA levels in infected THP1 and THP1-PDGFRα at 24, 72 and 120 hpi, measured by qPCR and normalized to a cellular genomic target. n = 2. **d** Nuclear viral genome counts in THP1 and THP1-PDGFRα monocytes at 12 hpi by DNA-FISH (n = 112 and n = 84, respectively). P-value was calculated using Poisson regression. **e** Quantification of viral particles within the cytoplasm of infected THP1 and THP1-PDGFRα monocytes using HCMV-UL32-GFP at 1 hpi (n = 78 and n = 62, respectively). Viral particles were counted using FIJI image processing, and statistics was performed using Poisson regression. **f** Proportion of new viral reads out of the total new reads detected by SLAM-seq in infected THP1 and THP1-PDGFRα monocytes. Infected cells were labeled with 4sU at 3 hpi and harvested for SLAM-seq after 2 or 3 h of labeling. **g** THP1 monocytes with constitutive PDGFRα expression, inducible PDGFRα expression or control cells, were infected with HCMV-GFP. Viral genomes in supernatants were quantified by qPCR at 8 dpi. Mean viral DNA levels relative to the control are presented. n = 3. p-value was calculated using a two-sided Student *t* test. **h** THP1 monocytes with constitutive PDGFRα expression, inducible PDGFRα expression or control cells were infected with HCMV-GFP. Supernatants harvested at 8 dpi were applied to wild-type fibroblasts, and GFP-positive cells were measured 48 h later by flow cytometry to determine PFU. Mean values are presented. n = 6. p-value was calculated using a two-sided Student *t* test. Source data for Fig. 4c–e, g, h are provided as a Source data file. Gating strategies for Fig. 4a, b are shown in Fig. S1a.

dox-inducible expression system in which PDGFRα expression is transiently induced prior to infection by addition of doxycycline (Fig. S4k). Also in these conditions, PDGFRα expression led to productive infection in a substantial percentage of cells (Fig. S4l). While in the constitutive system PDGFRα expression persists throughout infection, as expected, in the dox-inducible system the expression is transient and PDGFRα expression is diminished by 48 hpi (Fig. S4m). Viral particles are detected in the supernatant of both constitutive and inducible-PDGFRα infected cells (Fig. 4g). However, only in the inducible system, where PDGFRα levels decline before virus budding, as reported for fibroblasts[34], are the released particles infectious (Fig. 4h). Altogether, these results confirm that both transient and constitutive

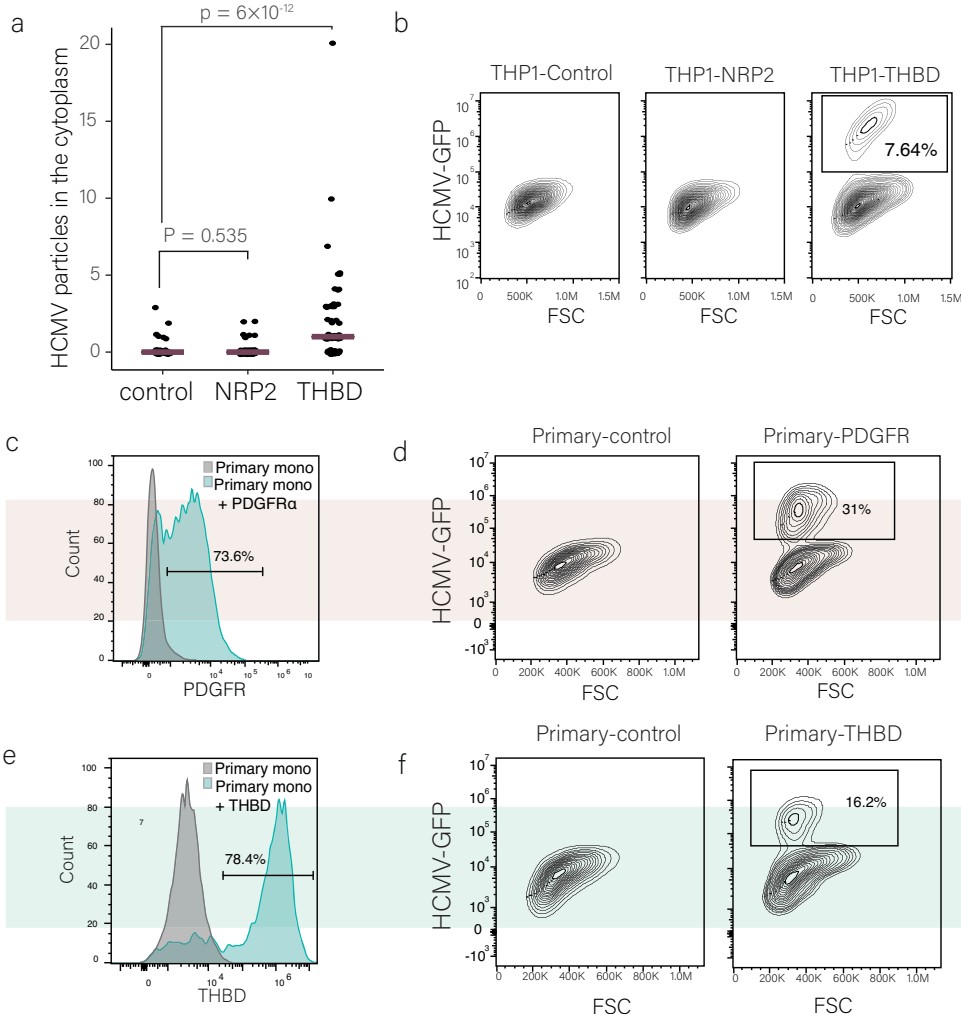

**Fig. 5 | PDGFRα and THBD expression facilitate productive infection in primary monocytes. a** Quantification of the number of viral particles, using HCMV-UL32-GFP, within the cytoplasm of infected THP1 monocytes with induced expression of mCherry as a control, NRP2 or THBD at 1 hpi (n = 50, 67, and 63, respectively). Viral particles were counted using FIJI image processing and statistical analysis was performed using Poisson regression. Source data are provided as a Source data file. **b** Flow cytometry analysis of THP1 monocytes overexpressing control, THBD and NRP2, using a dox-inducible system, infected with HCMV-GFP at 3 dpi. The gate marks the productive, GFP-bright cell population. **c, e** Flow cytometry analysis of PDGFRα surface expression on primary monocytes transfected with control, PDGFRα (**c**) and THBD (**e**) mRNA at 12 h after transfection. **d, f** Flow cytometry analysis of primary monocytes transfected with PDGFRα (**d**), THBD (**f**) or control mRNA for 12 h before HCMV infection. Cells were analyzed at 3 dpi. Gating strategies for Fig. 5b–f are shown in Fig. S1a.

PDGFRα expression increase HCMV entry into THP1 monocytes, supporting completion of the viral replication cycle, and suggest that constitutive PDGFRα expression likely interferes with particle infectivity[35].

## PDGFRa and THBD expression facilitate productive infection in primary monocytes

We next explored whether additional HCMV entry receptors can induce productive infection. Using the inducible system in THP1 monocytes, we over-expressed two additional established receptors of HCMV, NRP2 and THBD[30,36,37] (Fig. S5a), and then infected these cells with HCMV-GFP. While overexpression of NRP2 did not affect HCMV entry, overexpression of THBD led to a significant increase in viral entry into monocytes (Fig. 5a). Correspondingly, over-expression of THBD led to a distinct population of productively infected monocytes, as evident from high viral GFP expression as well as production of infectious viral progeny (Figs. 5b and S5b), while NRP2 overexpression did not affect infection (Fig. 5b). Productively infected cells were those with higher surface expression levels of

THBD suggesting a direct connection between efficient entry and the ability to establish productive infection (Fig. S5c). Interestingly, a smaller fraction of the THBD expressing cells were infected compared to infection with PDGFRα overexpression, suggesting differences in the extent of entry via these receptors. This is indeed supported by the extent of entry we measured (compare Figs. 4e and 5a). THBD overexpression in THP1 monocytes did not induce differentiation, as they did not differ from the parental THP1 cells morphologically (Fig. S5d) or in their expression levels of macrophage-related surface markers (Fig. S5e). Transcriptomic analysis revealed only minor changes in gene expression, all of which are not related to macrophage differentiation or innate immunity (Fig. S5f and Supplementary Data 4), negating the possibility of indirect effects of THBD expression.

To examine if this effect of enhancing viral entry can also be recapitulated in primary monocytes, we transfected primary monocytes with in vitro transcribed PDGFRα or control mRNAs (Fig. 5c). Infection of these cells with HCMV-GFP resulted in a distinct population of GFP-bright cells at 3 dpi only in the cells transfected with

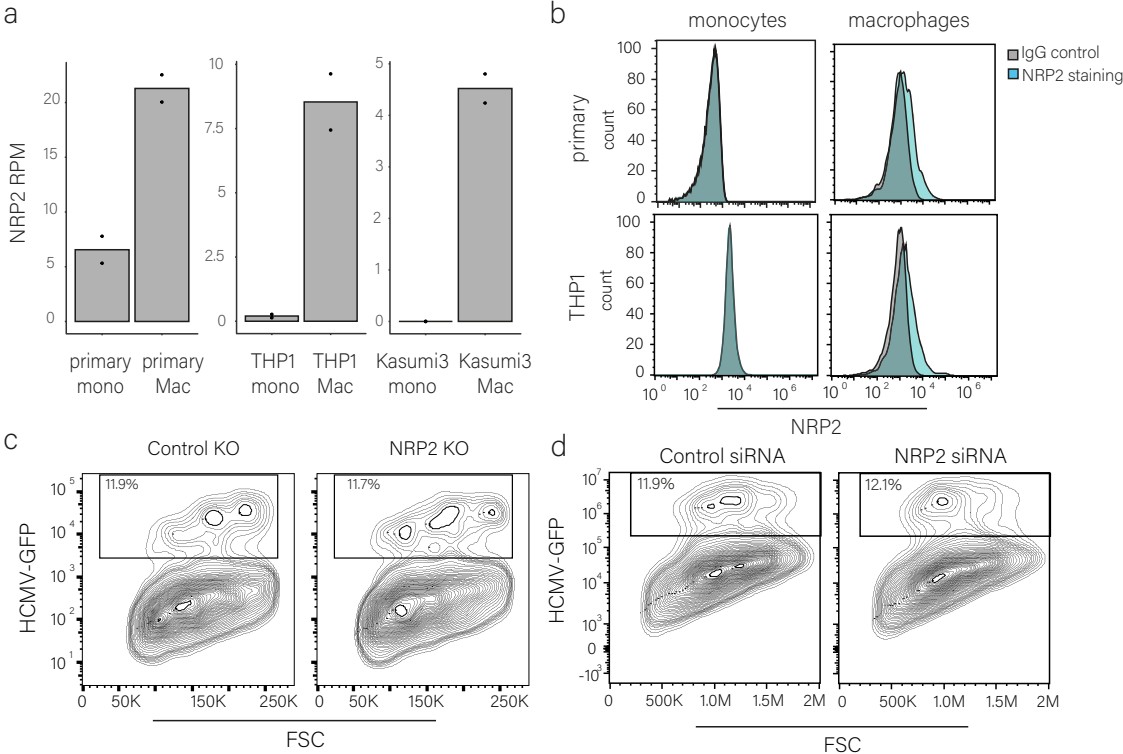

**Fig. 6 | NRP2 does not facilitate HCMV entry into macrophages. a** NRP2 expression in primary, THP-1 and Kasumi-3 monocytes and macrophages as measured by RNA-seq. Mono monocytes, Mac macrophages, RPM reads per million. n = 2. Source data are provided as a Source data file. **b** Flow cytometry analysis of NRP2 versus IgG control cell surface staining in primary and THP1 monocytes and macrophages. **c** Flow cytometry analysis of THP1 macrophages with NRP2 or control CRISPR knockout, infected with HCMV-GFP. Analysis was performed at 3 dpi. **d** Flow cytometry analysis of THP1 macrophages, transfected with NRP2 and control siRNA 2 days before infection with HCMV-GFP. Analysis was performed at 3 dpi. Gating strategies for Fig. 6b–d are shown in Fig. S1a.

PDGFRα mRNA (Fig. 5d). Although extensive cell death (likely resulting from transfection-related stress) prevented analysis of viral progeny, we detected robust viral genome replication (Fig. S5g). Similarly, transfection of primary monocytes with THBD mRNA (Fig. 5e) resulted in a distinct population of GFP-bright cells (Fig. 5f). It is noteworthy that the effect of ectopic expression on infection in primary cells was smaller than that observed in THP1 cells. These differences may reflect differences in entry per se, for example, due to the presence of a co-factor, or may reflect additional barriers in primary monocytes, beyond entry. Regardless, these results indicate that increasing viral entry in primary monocytes facilitates initiation of viral gene expression and viral genome replication indicating productive infection is likely taking place.

**Integrin β3 plays a role in HCMV entry into macrophages**
Our findings indicate that differences in viral entry efficiency contribute to the different outcomes of HCMV infection in monocytes and macrophages, but the source of these differences in entry remains unclear. We analyzed the potential involvement of several cell surface receptors implicated in HCMV infection, which are upregulated upon differentiation. We first focused on NRP2, whose transcript levels increase upon differentiation of monocytes to macrophages in the three cell types tested (Fig. 6a). Indeed, cell surface staining illustrated NRP2 is not expressed on the surface of monocytes and differentiation to macrophages was accompanied by low but detectable cell surface expression (Fig. 6b). We found that NRP2 ectopic expression in monocytes is not sufficient for inducing productive infection in monocytes (Fig. 5b), however this could be due to the absence of additional factors which act together with NRP2 to facilitate entry. To test whether NRP2 is necessary for viral entry into macrophages, we

used CRISPR-Cas9 to generate THP1 cells in which NRP2 is knocked out, which resulted in a partial knockout (Fig. S6a). We therefore also used siRNA to knockdown NRP2 expression (Fig. S6b). Infection of differentiated cells in which NRP2 was knocked out or down showed similar levels of infection compared to control cells (Figs. 6c, d and S6c, d), indicating NRP2 likely does not mediate HCMV entry into macrophages. To verify that our results are not impacted by possible mutations acquired during viral propagation, we sequenced the virus we used for infection (TB40 strain) and ruled out accumulation of mutations in the genes encoding the viral entry receptors (see "Methods" section).

We also analyzed two integrins, ITGB3 and ITGB1, which encode for integrin β3 and β1, respectively, both were significantly upregulated upon differentiation (Fig. 7a). These integrin β subunits can dimerize with different α subunits to form canonical heterodimers, some of which were implicated in HCMV entry[14,38,39]. In agreement with our RNA-seq measurements, β3 surface expression was not detected in either primary or THP1 monocytes, whereas in macrophages its expression was pronounced (Fig. 7b); β1 was expressed on the surface of monocytes but its expression significantly increased in macrophages (Figs. 7c and S7a).

To dissect if these integrins play a role in HCMV entry into macrophages, we generated CRISPR knockouts of either ITGB3 or ITGB1 in THP1 cells (Fig. S7b). Knockout of ITGB1 did not affect HCMV productive infection in THP1-derived macrophages. However, in the absence of ITGB3, differentiated macrophages were much less susceptible to productive infection compared to control cells (Figs. 7d and S6c). We also tested the knockout effect of both ITGB3 and ITGB1 but observed no cumulative effect beyond the effect of ITGB3 knockout (Figs. 7d and S7c).

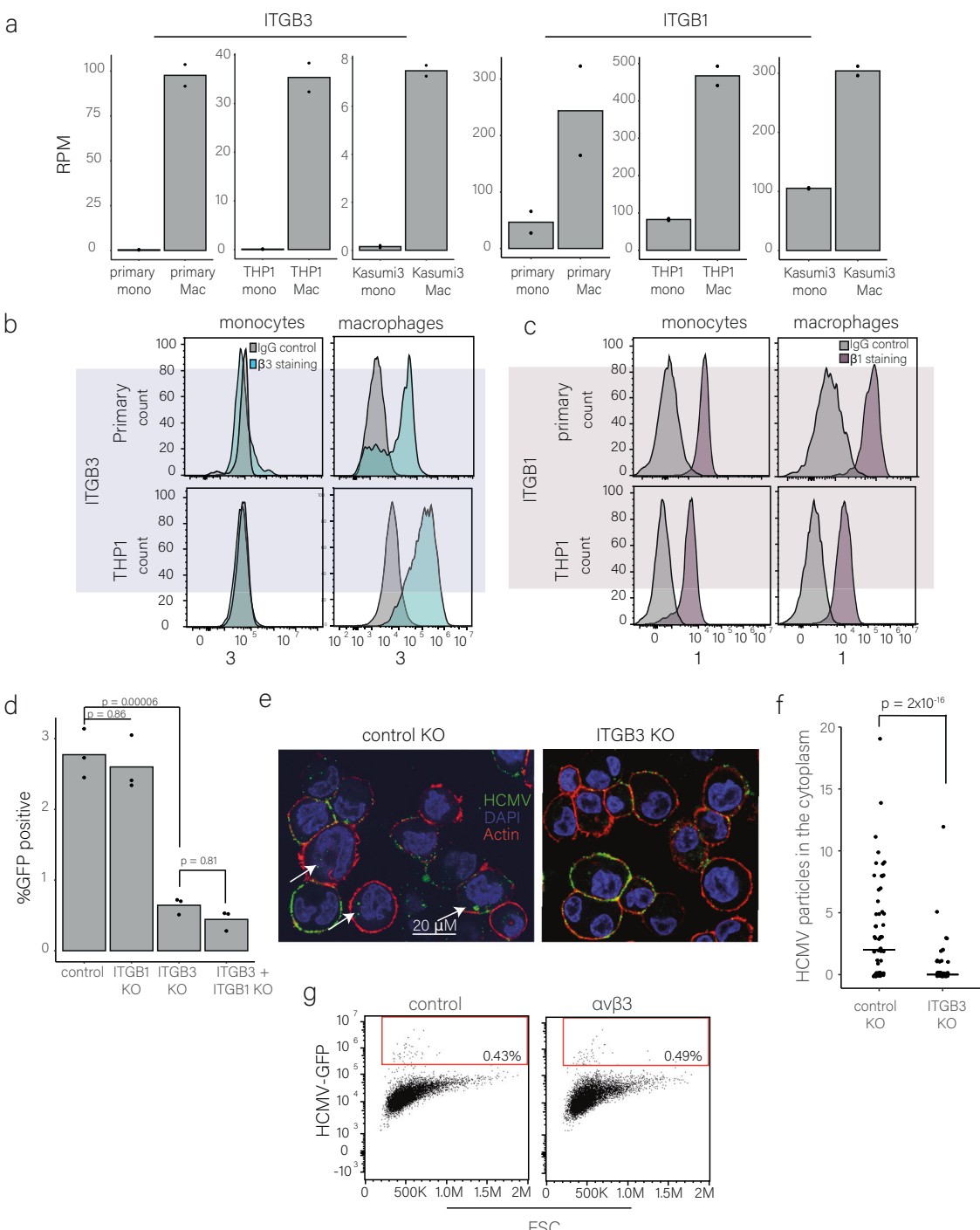

**Fig. 7 | HCMV entry into macrophages is mediated through ITGB3. a** ITGB3 and ITGB1 expression in primary, THP1 and Kasumi-3 monocytes and macrophages as measured by RNA-seq. RPM, reads per million. n = 2. **b, c** Flow cytometry analysis of integrin β1 (**b**) and integrin β3 (**c**) cell surface levels in primary and THP1 monocytes and macrophages. **d** Flow cytometry analysis of control, ITGB3, ITGB1 and ITGB3 + ITGB1 knockout (KO) in THP1 macrophages infected with HCMV-GFP. Cells were analyzed at 3 dpi, p-value was calculated using a two-sided Student *t* test. n = 3. **e, f** Representative microscopy images (**e**) and quantification of viral particles using FIJI analysis software (**f**) of THP1 macrophages with ITGB3 knockout (KO) versus control knockout (n = 67 and n = 60, respectively) infected with HCMV-UL32-GFP and imaged at 1 hpi Actin staining was used to visualize the cells' borders and DAPI for nuclei staining. Statistics was performed using Poisson regression. **g** Flow cytometry analysis of infected THP1 monocytes overexpressing ITGB3 and ITGAV (αvβ3), compared to mCherry control. Overexpression was induced for 24 h using doxycycline prior to HCMV-GFP infection. Cells were analyzed at 3 dpi. Gating strategies for Fig. 7b, c, d, g are shown in Fig. S1a. Source data for Fig. 7a, d, f are provided as a Source data file.

We further validated these results by performing siRNA knockdown of ITGB3 in THP1 macrophages (Fig. S7d), showing that also knockdown leads to a significant decrease in productive infection of macrophages (Fig. S7e, f). By infecting with a UL32-GFP virus, we further show that ITGB3 KO in macrophages significantly reduced viral entry (Fig. 7e, f). Together, our results illustrate that the expression of ITGB3 significantly increases upon monocyte to macrophage differentiation and that this increase plays a role in facilitating HCMV entry and subsequently in promoting productive infection of macrophages.

We next tested whether ectopic expression of ITGB3 alone is sufficient to promote productive infection in monocytes. However, ITGB3 overexpression did not enhance infection compared to control cells (Fig. S7g, h). Given that ITGB3 functions in complex with ITGAV (integrin αV), which is also upregulated during monocyte differentiation (Supplementary Data 2), we co-expressed ITGB3 and ITGAV in THP1 monocytes using an inducible promoter (Fig. S7i, j). After sorting for double-positive cells (Fig. S7j), we infected them with HCMV-GFP. Co-expression of ITGB3 and ITGAV (αVβ3) likewise failed to increase productive infection relative to control cells (Figs. 7g and S7k). To verify the functionality of ectopically expressed ITGB3, we complemented ITGB3 KO and control macrophages by overexpressing ITGB3 (with synonymous mutation to allow resistance to CRISPR editing) and we show that indeed complementation occurs and infection is restored, and moreover, overexpression of ITGB3 in macrophages further increases productive infection (Fig. S7l). These results demonstrate that, although ITGB3 plays a role in HCMV entry into macrophages, its increased expression in macrophages compared to monocytes is not the sole factor that facilitates the changes in viral entry and other proteins that are induced upon differentiation are likely required.

### Latency is established in cells receiving lower viral load

Our results suggest a hypothesis by which the number of incoming viral genomes plays a major role in determining the likelihood of productive infection. When the number of incoming viral genomes is low, productive infection is unlikely and instead repression and latent infection may ensue. To test this, we aimed to isolate infected cells with lower load of viral genomes, which we hypothesized corresponds to infected cells that fail to initiate immediate early viral gene expression. We then sought to determine whether these cells contain latently infected cells (Fig. 8a). To directly assess this hypothesis, we infected THP1 monocytes carrying an inducible PDGFRα with a triple fluorescent HCMV strain carrying fluorescent tags for immediate early (IE), early and late viral gene expression[40]. At 16 hpi, prior to any viral genome replication, we sorted cells according to their IE expression to bright and dim populations (Fig. 8b). We found that the viral genome levels were >15-fold lower in dim compared to bright cells (Fig. 8c), indicating that the dim cells are the cells that initially received fewer viral genomes. The dim population was re-sorted at 5 dpi to ensure there were no lytic cells, which were not detected at 16 hpi. We followed the two cell populations and found that at 7 dpi, the sorted IE-bright cells are indeed lytically infected, as they robustly expressed the late viral gene marker and produced infectious viral progeny while the IE-dim cells did not (Figs. 8d and S8a). Importantly, the isolated dim cells were capable of reactivation and release of infectious viral progeny upon differentiation at 7 dpi (Figs. 8e and S8b), indicating they were latently infected, while the release of infectious viral progeny from bright cells was initially high and unaffected by differentiation (Fig. S8c). These data show that indeed cells that receive fewer viral genomes have a lower chance of becoming productively infected and are the ones in which latency is established.

To further substantiate these results, we infected THP1-PDGFRα with HCMV-GFP at different MOIs and sorted the dim cells at 5dpi (Fig. 8f). As expected, with increasing MOI the percentage of lytic cells increased. In all MOIs the viral load was significantly lower in the dim cells sorted at 5dpi compared to the viral load in all cells at the early stage of infection (Fig. 8g), indicating that these cells represent the cells that initially received fewer viral genomes and illustrating that the number of particles that infect a cell plays a major role in dictating infection outcome in monocytes. Across all MOIs tested, these dim cells retained the capacity of reactivation upon differentiation (Fig. S8d), indicating at least some cells were latently infected. These results propose a model by which the number of incoming viral genomes determines the likelihood of productive infection. When the

number of viral genomes per cell falls below a critical threshold, the chances of viral genome repression increases, and latency can be established (Fig. 9).

Overall, these results demonstrate that inefficient entry of HCMV into monocytes is a major factor underlying the low levels of viral gene expression and, consequently, the ability to establish latency. Facilitating efficient entry of viral genomes enables productive replication even in monocytes, underscoring that entry is a critical determinant of infection outcome in these cells.

## Discussion

HCMV infection of monocytes results in a latent infection in which the virus is largely repressed and does not replicate, while following differentiation of these cells, they become permissive to productive infection and produce progeny.

Previous studies have shown that latent infection is characterized by low levels of viral transcripts[41,42]. Using metabolic labeling, we show that viral gene transcription is very low in monocytes while substantial in macrophages. Chromatin regulation has been implicated as a major factor for viral repression during latency[11,13]. Although some specific chromatin factors have been attributed to this specific repression[43–45], the differences between monocytes and their differentiated counterparts with regard to chromatin repression remain poorly defined. Our findings indicate that chromatin-based repression does not fully account for the differences between these cell types. Using HDAC inhibitors, chromatin repression can be relieved and indeed results in elevated productive infection in both monocytes and macrophages, but still productive infection in monocytes remains very limited.

We found a striking disparity in the abundance of viral genomes within the nuclei of infected monocytes compared to macrophages at early time points, in line with previous studies of monocyte infection[46], and notably, a significant difference in the quantity of viral capsids within these cells. This suggests HCMV entry efficiency as an unexplored barrier for productive infection in monocytes. Remarkably, ectopic expression of known HCMV entry receptors in monocytes facilitates productive infection, underscoring inefficient entry as a barrier in these cells. This means that although monocytes are capable of supporting productive infection, they do not reach a certain threshold of viral genome load required to establish productive infection due to less efficient viral entry in these cells (Fig. 9).

By performing unbiased transcriptome analyses, we interestingly found significant upregulation upon differentiation of several cell surface proteins, which are linked to the entry of HCMV. One of these receptors was NRP2, which mediates HCMV entry into non-fibroblast cells, through the viral pentamer complex. Although its expression increased during differentiation, our results indicate that it does not play a major role in viral entry into macrophages, at least with the TB40 strain. ITGB3 and ITGB1 were both shown to play a role in HCMV entry[14,38,47] and depletion of these integrins showed that ITGB3, which is not expressed in monocytes, is required for HCMV entry into macrophages. However, overexpression of ITGB3 alone or with its canonical partner, ITGAV, was not sufficient to enable productive infection in monocytes, indicating the involvement of additional factors or post-translational modifications[48] that are required for entry and likely absent or low in monocytes.

The notion that entry constitutes a major barrier for productive infection in monocytes suggests that the number of particles that infect a cell plays a major role in the probability of establishing productive versus latent infection. The finding that in infected monocytes, cells receiving fewer viral genomes fail to establish productive infection and instead enter latency further supports this conclusion. Therefore, we suggest a model by which when the number of viral particles in a cell falls below a critical threshold, productive infection does not occur, likely due to the inability to prevent repression of the genome by the host, and instead latent infection may arise (Fig. 9). We

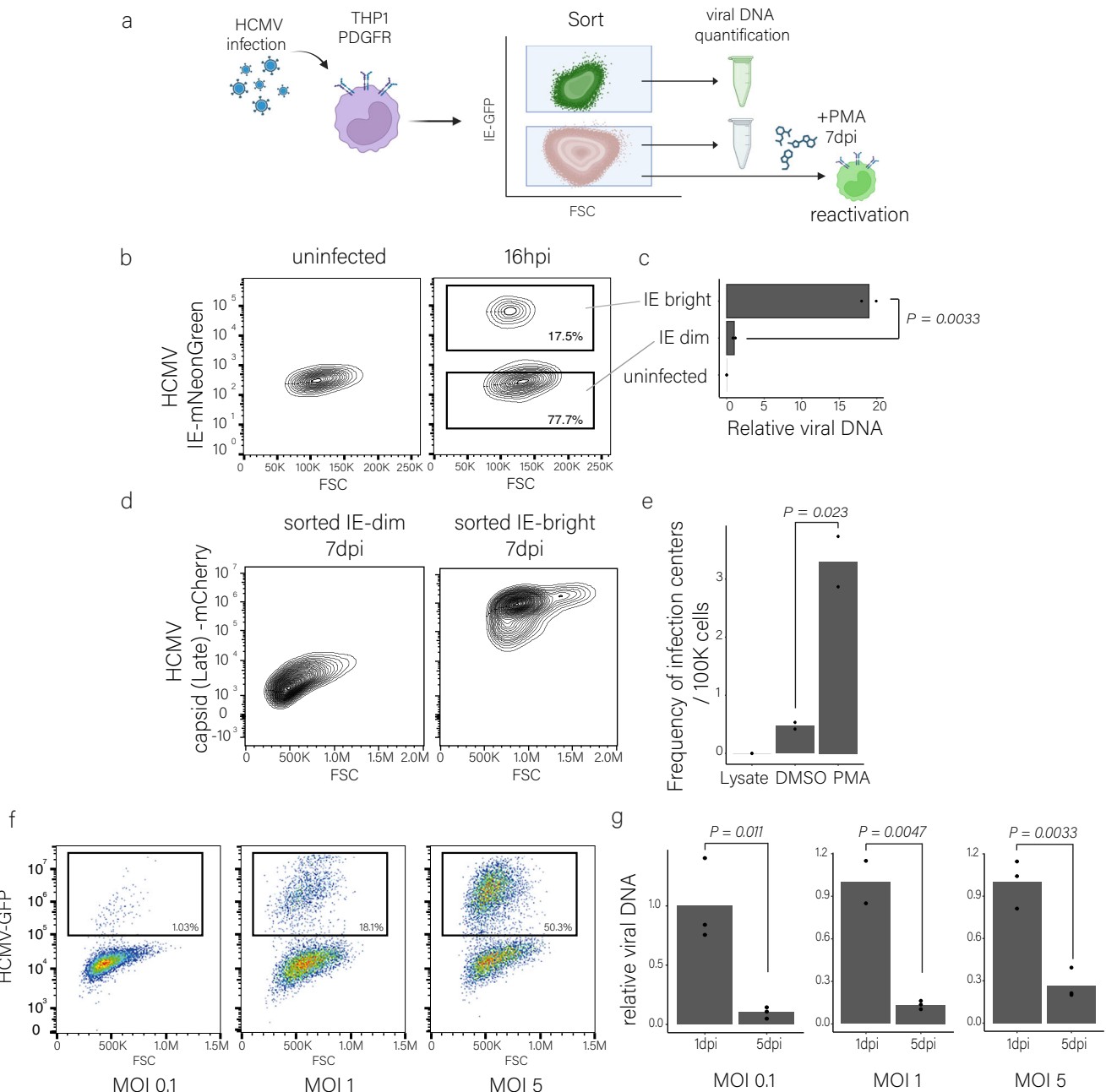

**Fig. 8 | THP1-PDGFRa receiving fewer viral genomes establish latent infection.**
**a** Illustration of the experimental setup. Created in BioRender. Schwartz, M. (2025)
https://BioRender.com/uwraqo4. **b** THP1 monocytes were induced to express
PDGFRα and were infected with a triple fluorescent HCMV strain carrying fluor-
escent tags for immediate early (IE), early and late viral gene expression[40]. Cells
were sorted at 16 hpi based on mNeonGreen expression from the IE promoter to
bright and dim populations. **c** Relative viral DNA levels in sorted bright and dim
populations at 16 hpi. Viral DNA levels were measured by qPCR and were normal-
ized to a cellular genomic target. p-value was calculated using a two-sided Student *t*
test. n = 3. **d** Flow cytometry analysis at 7dpi of mCherry from the viral late gene
promoter (UL48A) in HCMV-mNeonGreen (IE)-bright and dim sorted cells.

**e** Reactivation levels of HCMV in sorted THP1−PDGFRα dim cells measured on
fibroblasts as frequency of infectious centers calculated with ELDA software[50,51].
Undifferentiated cells and cell lysates from 7 dpi are shown as controls. p-value was
calculated using a two-sided Student *t* test. n = 2. **f** Flow cytometry analysis of
THP1−PDGFRα infected with HCMV-GFP at MOIs of 0.1, 1 and 5. The dim population
was sorted according to GFP expression at 5 dpi. **g** Relative viral DNA levels from
THP1−PDGFRα cells at 1 dpi and from sorted dim cells at 5 dpi, measured from
extracted DNA. Viral DNA levels were measured by qPCR and were normalized to a
cellular genomic target. p-value was calculated using a two-sided Student *t* test.
n = 3 except in 1 dpi, MOI = 1, where n = 2. Gating strategies for Fig. 8b, d, f are shown
in Fig. S1a. Source data for Fig. 8c, e, g are provided as a Source data file.

show here that the composition of receptors on the cell surface affects
entry in the case of monocyte to macrophage differentiation, however
additional factors may affect viral genome entry, such as cell state,
exposure to interferons, etc. Moreover, different cell types may have
different thresholds for the amount of particles required for estab-
lishing productive infection. This may be affected for instance by the
balance of activating versus repressing chromatin profiles. This raises

the possibility that infection at low multiplicity, possibly in diverse cell
types, may lead to a subset of cells harboring an insufficient amount of
genomes or particles to support productive infection, these will likely
be repressed and potentially still facilitate the establishment of
latency.

Altogether, our findings identify inefficient entry as a critical
barrier to productive infection in monocytes, underscoring an

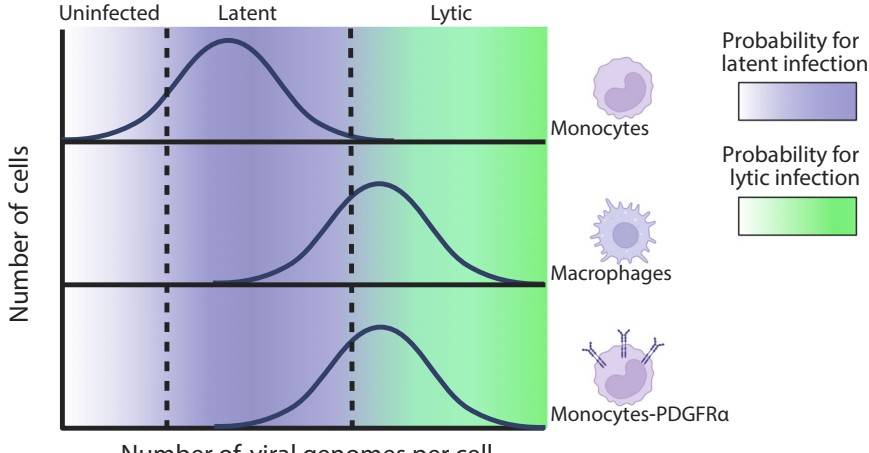

**Fig. 9 | Model for the effect of viral entry on productive versus latent HCMV infection.** Schematic model illustrating the relationship between viral genome load and infection outcome. The histograms represent the distribution of viral genomes per cell within the population. The color gradients indicate the probability for latent infection (purple) or lytic infection (green), with dashed lines marking the approximate thresholds. Monocytes typically receive fewer viral genomes, favoring latency, compared to macrophages. Ectopic expression of an HCMV receptor, e.g., PDGFRα, in monocytes, increases the viral genome load driving them towards lytic infection. Created in BioRender. Schwartz, M. (2025) https://BioRender.com/x3v09b4.

overlooked yet pivotal aspect of monocyte susceptibility to latent HCMV infection.

## Methods

### Ethics statement

All fresh peripheral blood samples were obtained after approval of protocols by the Weizmann Institutional Review Board (IRB application 92-1) and following informed consent from the donors. Blood donors were not compensated.

### Cell culture and virus

Primary CD14+ monocytes were isolated from fresh venous blood, obtained from healthy donors, males and females, aged 25–45, using Lymphoprep density gradient (Stemcell Technologies) according to the manufacturer instructions, followed by magnetic cell sorting with CD14+ magnetic beads (Miltenyi Biotec) according to the manufacturer instructions. Monocytes were cultured in X-Vivo15 media (Lonza) supplemented with 2.25 mM L-glutamine at 37 °C in 5% CO$_2$, at a concentration of 1–2 million cells per ml in non-stick tubes to avoid differentiation.

Where indicated, primary monocytes were differentiated immediately following isolation by culturing in RPMI with 20% heat-inactivated fetal bovine serum (FBS), 2 mM L-glutamine and 100 units ml$^{-1}$ penicillin and streptomycin (Beit-Haemek) supplemented with 50 ng/ml PMA (Sigma) for 3 days. Culturing of monocyte derived macrophages was in RPMI with 20% heat-inactivated fetal bovine serum (FBS), 2 mM L-glutamine and 100 units ml$^{-1}$ penicillin and streptomycin (Beit-Haemek).

293 T cells (ATCC CRL-3216) and Primary human foreskin fibroblasts (ATCC CRL-1634) were maintained in DMEM with 10% FBS, 2 mM L-glutamine and 100 units ml$^{-1}$ penicillin and streptomycin (Beit-Haemek).

THP1 cells, purchased from ATCC (TIB-202), and Kasumi-3 cells, purchased from ATCC (CRL-2725), were grown in RPMI media with 20% heat-inactivated FBS, 2 mM L-glutamine and 100 units/ml penicillin and streptomycin (Beit-Haemek). Differentiation of THP1 and Kasumi-3 cells was done by adding 50 ng/ml PMA for 3 days.

Twenty-four hours before infection, cells were grown in 0.5% or 10% FBS For experiments in which the cells were collected at 3dpi or progeny, respectively, and supplemented with 10% FBS after 1 h of infection.

In the experiments in which receptors were induced before infection, cells were treated with 1 μg/ml doxycycline, with or without 1 μM TrichostatinA (TSA, Sigma) for 24 h before infection. The cells were validated for the receptor expression by surface staining and washed before HCMV infection.

TSA was added at the specified concentrations, at 5 hpi, DMSO was added as control.

IFNγ (500U/ml, Peprotech) was added for 2 days to THP1 monocytes before surface staining of MHC-II.

The TB40E strain of HCMV, containing an SV40–GFP reporter, was described previously[15]. For microscopy imaging, a previously reported TB40E strain containing GFP fused to a tegument protein (UL32-GFP) was used[32]. For the reactivation experiment presented in Fig. 8b–e, a previously described TB40 strain expressing triple fluorescence was used[40].

Virus propagation was done by adenofection of a bacterial artificial chromosome of the viral genome into fibroblasts as described previously[49]. When most of the cells in the culture died, the supernatant was collected and cleared from cell debris by centrifugation.

### Infection procedures

Infection was performed by centrifugation at 800 × g for 1 h in 24-well or 12-well plates with the virus added at multiplicity of infection (MOI) = 5, unless stated differently, followed by washing and supplementing with fresh media. Notably, because this MOI is based on quantification of infectious particles in fibroblasts it is effectively lower in monocytic cells.

For progeny assay, at 8 or 10 dpi. The supernatant was cleared from cell debris by centrifugation and transferred to fibroblasts. Infected fibroblasts were counted 2–3 days later.

For Reactivation assays described in Figs. S1b and S8d, GFP dim cells were sorted at the indicated time point, as specified (in monocytes with no PDGFRα expression we validated there are no GFP-bright cells) and at 7 dpi were treated with PMA (50 ng/ml) or with DMSO as control. HCMV-positive cells were counted on a fluorescent microscope. Reactivation assay using Extreme Limiting Dilution Assay (ELDA), described in Fig. 8e, was performed as previously described[50] at 7 dpi, cells were seeded with PMA or with DMSO, as control, in serial dilutions ranging from 20,000 cells to 625 cells per well in 96 well plate, with 8 replicates per dilution. The equivalent number of cells were lysed and seeded to control for infectious virus in the initial cells.

After 2 days, 10,000 fibroblasts were added to each well to allow detection of infectious virus. At 24 dpi, plaques were counted, and reactivation levels were quantified using an ELDA analysis webtool[51].

## Flow cytometry and sorting

Adherent cells were harvested by washing with PBS and in 0.5 mM EDTA before scraping. Cells were analyzed on a BD Accuri C6 or CytoFLEX (Beckman Coulter) and sorted on a BD FACS AriaIII using FACSDiva software. All analyses and figures were done with FlowJo. All histograms were plotted with modal normalization.

## Next generation sequencing

RNA-seq library preparation was performed as described previously[52]. Cells were collected with Tri-Reagent (Sigma-Aldrich). Total RNA was extracted according to the manufacturer instructions and poly-A selection was performed using Dynabeads mRNA DIRECT Purification Kit (Invitrogen). The mRNA samples were subjected to DNase I treatment and 3′ dephosphorylation using FastAP Thermosensitive Alkaline Phosphatase (Thermo Scientific) and T4 polynucleotide kinase (New England Biolabs) followed by 3′ adaptor ligation using T4 ligase (New England Biolabs). The ligated products were used for reverse transcription with SSIII (Invitrogen) for first-strand cDNA synthesis. The cDNA products were 3′-ligated with a second adaptor using T4 ligase and amplified for 8 cycles in a PCR for final library products of 200–300 bp. Raw sequences were first trimmed at their 3′ end, removing the Illumina adapter and poly (A) tail. Alignment was performed using Bowtie 1[53] (allowing up to two mismatches) and reads were aligned to the human reference (hg19). Reads aligned to ribosomal RNA were removed. Reads that were not aligned to the genome were then aligned to the transcriptome.

For sequencing the genome of the HCMV-GFP virus used for infection, DNA was extracted from infected THP1-PDGFRα cells using blood kit (QIAGENE). The sequencing library was prepared using NEBNext® DNA Library Prep Kit according to the manufacturer instructions. Reads were aligned to the TB40 reference genome (EF999921) using STAR. The resulting wiggle file was inspected manually, and no mutations or deletions were detected in the genes encoding the viral entry receptors.

## Differential expression and enrichment analysis

Differential expression analysis on RNA-seq data was performed with DESeq2 (v.1.22.2) using default parameters, with the number of reads in each of the samples as an input. The log2(fold change) values from the DE on the RNA-seq was used for enrichment analysis using GSEA (v.4.1). For the analysis, only genes with a minimum of ten reads were included.

Gene lists of cellular receptors[30], chromatinization[54] and ISGs[55] are based on the referred papers. Monocyte to macrophage differentiation gene list is based on the list of genes in the GO term regulation of macrophage differentiation on GO:0030225.

## RNA labeling for SLAM-seq and analysis

For metabolic RNA labeling, 4sU (T4509, Sigma) was added at a final concentration of 200 µM to infected cells at 3 hpi. Cells were collected with Tri-reagent at 2 and 3 h after adding the 4sU (corresponding to 5 and 6 hpi). RNA was extracted under reducing conditions and treated with iodoacetamide (A3221, Sigma) as previously described[18]. RNA-seq libraries were prepared and sequenced as described in the "Next-generation sequencing" section.

Alignment of SLAM-seq reads was performed using STAR, with parameters that were previously described[56]. First, reads were aligned to a reference containing human rRNA and tRNA, and all reads that were successfully aligned were filtered out. The remaining reads were aligned to a reference of the human (hg19) and the TB40 (EF999921). In one analysis, the virus was analyzed as one transcript, and in a second analysis, all viral genes were analyzed. Output.bam files from STAR were used as input for the GRAND-SLAM analysis[57] with default parameters and with trimming of 5 nucleotides in the 5′ and 3′ ends of each read. Each one of the runs also included an unlabeled sample (no 4sU) that was used for estimating the linear model of the background mutations. The estimated ratio of newly synthesized out of total molecules for each viral and host gene were used for the presented analyses.

## 3D immunofluorescence viral DNA FISH

Differentiated monocytes (THP1 and CD14+) were seeded on 22 × 22 coverslips in a 6-well plate. Suspension monocytes were concentrated to 1 M cells/150 µl and seeded on 22 × 22 coverslips for 1 h, followed by centrifuging for 10 min in 800 × g to deposit suspension cells onto the coverslips. FISH was done as previously described[58]. Cells were washed twice with PBS and fixed in 4% paraformaldehyde in PBS for 10 min. Then, cells were permeabilized in 0.5% Triton/PBS for 15 min and rinsed in PBS. Samples were incubated in 20% glycerol/PBS at 4 °C overnight and frozen five times in liquid nitrogen. Cells were washed three times in 0.05% Triton/PBS, rinsed with PBS and with DDW. Cells were then incubated in 0.1 M HCL for 15 min and then with 0.002% pepsin (Sigma, P6887)/0.01 M HCl at 37 °C for 90 s for monocytes or 75 s for macrophages, followed by inactivation in 50 mM Mgcl$_2$/PBS. Cells were then fixed in 1% paraformaldehyde/PBS for 1 min and washed with PBS and with 2X SSC (Promega, V4261), followed by Incubation in 50%formamide/2X SSC (pH7–7.5) for 1 h.

Probe was prepared from a TB40 SV40-GFP BAC DNA using a Nick Translation Mix (Roche, 11745808910, Tetramethyl-Rhodamine-5-dUTP, Roche 11534378910) according to the manufacturer instructions, and prepared for hybridization by combining 0.5 mg of each labeled probe and 5 mg cot-1 (Invitrogen, 15279-011) in 4.5 µl deionized formamide (Sigma, F9037) and 4.5 µl 4XSSC/20% dextran sulfate.

Following denaturation at 76 °C, hybridization was performed at 37 °C for 3 days in a humid chamber. After hybridization, the coverslips were washed in 50% formamide/2X SSC (pH 7–7.5) at 37 °C, in 0.5XSSC at 60 °C and in 4XSSC/0.2%. Coverslips were mounted on slides with Prolong gold (Invitrogen, P36930) containing DAPI.

## Immunofluorescence

For HCMV replication compartment detection, cells were plated on i-bidi slides, fixed in 4% paraformaldehyde for 15 min, washed in PBS, permeabilized with 0.1% Triton X-100 in PBS for 10 min and then blocked with 10% goat serum in PBS for 30 min. Immunostaining was performed with mouse anti-UL44 (CA006-100, Virusys) diluted in 2% goat serum diluted in PBS. Cells were washed 3 times with PBS and labeled with goat anti-mouse–Alexa Fluor 647 (Thermo Fisher) secondary antibody and DAPI (4′,6-diamidino-2-phenylindole) diluted 1:500 in PBS for 1 h at room temperature, followed by 3 PBS washes.

Detection of the HCMV tegument protein (pp150, UL32) in the cytosol was performed by infecting the cells with the TB40-UL32-EGFP[32] for 1 h, followed by three PBS washes and 15 min of fixation with 4% paraformaldehyde. Samples were mounted with DAPI for nuclear staining and Phalloidin, which binds F-actin, to define cell borders.

## Microscopy and image analysis

DNA-FISH and HCMV capsid images were taken using Leica TCS SP8 CLSM. DNA-FISH images were analyzed using Imaris 10 software. Image files were anonymized by renaming and mixing to ensure blinded analysis. Positive signals were manually counted for each cell.

Infected cells with fluorescently labeled HCMV tegument protein (UL32-GFP) were visualized and analyzed using ImageJ (FIJI, NIH). Automated segmentation of individual virions was performed using the StarDist plugin[59], while cell boundaries were manually delineated to define individual cell regions of interest (ROIs). Virions located

within each cell ROI were assigned accordingly, enabling quantification of viral entry at single-cell resolution.

Images of HCMV replication compartment, bright field images and images of GFP positive infected cells were acquired on an AxioObserver Z1 wide-field microscope.

## Cell surface staining

Cells were washed three times with PBS and blocked for 15 min in 2% human serum. After blocking, cell staining was done using the following conjugated antibodies: allophycocyanin (APC)-conjugated mouse IgG2a anti-human neuropilin-2 (R&D systems, catalog no. FAB22151A) with allophycocyanin (APC)-conjugated mouse IgG2a control (R&D systems, catalog no. IC003A). Phycoerythrin (PE)-conjugated mouse IgG2a anti-human PDGFRα (BD, catalog no. 556002) with phycoerythrin (PE)-conjugated mouse IgG2a control (BD, catalog no. 555574). Alexa fluor 647 conjugated mouse IgG1 anti-human CD61 (ITGB3) (BLG, catalog no. 336407) and Alexa fluor 647 conjugated mouse IgG1 control (BLG, catalog no. 400130). FITC-conjugated mouse IgG1 anti-human CD29 (ITGB1) (Santa Cruz catalog no. MEM-101A) and FITC-conjugated mouse IgG1 control (Santa Cruz, catalog no. sc-2339). APC-conjugated Rat IgG2b, k anti-human CD11b (BioLegend catalog no. 101211). APC-conjugated Rat IgG1, k anti-human CD115 (BioLegend catalog no. 347305) and PE-conjugated mouse IgG2b anti-human CD11c (BD catalog no. 333149). APC-conjugated anti-HLA-DR mouse IgG2aκ (130-113-963).

Antibody incubation was done for 30 min at a 1:200 dilution. After staining, cells were washed twice with PBS and analyzed by flow cytometry.

## Plasmid construction lentiviral transduction

PDGFRα was cloned into pLex_TRC206 plasmid[60] under EF-1α promoter and blasticidin selection. PDGFRα was amplified from cDNA (Supplementary Data 5) and cloned into pLex by FastCloning[61].

For CRISPR knockout plasmids of NRP2, ITGB3 and ITGB1, we cloned two gRNAs into the lentiCRISPRv2-2guide plasmid using restriction-free cloning as previously described[62] (Supplementary Data 5). Trip10 was used as a knockout control as described in ref. 62.

To generate inducible expression plasmids of THBD, NRP2, ITGB3 and ITGAV, genes were ordered from TWIST (NCBI ID are 7056, 8828, 3690 and 3688, respectively). PDGFRa was amplified from the pLEX-PDGFRα plasmid. The genes were cloned into pLVX-Puro-TetONE-SARS-CoV-2-nsp1-2XStrep (kind gift from N. Krogan, UCSF) in place of the SARS-CoV-2-nsp1-2XStrep cassette using linearization with BamHI and EcoRI (neb). The genes were amplified with primers containing flanking regions homologous to the vector (Supplementary Data 5). The amplified PCR fragments were cleaned using a gel extraction kit (QIAGEN) according to the manufacturer's protocol and were cloned into the vectors using a Gibson assembly reaction (neb). Inducible mCherry, cloned on the same backbone was previously described[10] and was used as control in all experiments with these inducible expression plasmids.

Lentiviral particles were generated by cotransfection of the expression constructs and second-generation packaging plasmids (psPAX2, Addgene, catalog no. 12260 and pMD2.G, Addgene, catalog no. 12259), using jetPEI DNA transfection reagent (Polyplus transfection) into 293T cells, according to the manufacturer's instructions. At 60 h post-transfection, supernatants were collected and filtered through a 0.45-μm polyvinylidene difluoride filter (Millex). THP1 cells were transduced with lentiviral particles by centrifugation at $800 \times g$ for 1 h in 24-well or 12-well plates. Two days post transfection the cells were transferred to selection media (blasticidin, 10 μg/ml for 5 days or puromycin, 1.75 μg/ml for 4 days). Blasticidin and puromycin were removed and cells were recovered for at least two days before subsequent processing.

## CRISPR and siRNA-mediated knockdown

CRISPR-mediated knockout of NRP2 and ITGB1 was performed in bulk populations. For ITGB3, knockout was performed by CRISPR, followed by clonal selection of a single-cell-derived population and confirmation of mutations in both alleles by sequencing. In parallel, a control knockout of TRIP10 was generated in a bulk population for comparison.

siRNA-mediated knockdown of NRP2 and ITGB3 was conducted using ONTARGETplus SMARTpool reagents (Dharmacon; L-017721-00-0005 for NRP2 and L-004124-00-0005 for ITGB3), with a non-targeting control (D-001810-10-05). siRNAs were first diluted to 5 μM in siRNA buffer, then further diluted to 0.25 μM in Opti-MEM to prepare Solution I. In parallel, 4 μL of Lipofectamine RNAiMAX (Thermo Fisher) was diluted in 96 μL Opti-MEM to generate Solution II. Solutions I and II mixed together and were incubated for 20 min at room temperature.

The transfection mixture was added to THP1-derived macrophages at 1 day post-differentiation, in RPMI medium supplemented with 50 ng/mL PMA, to a final volume of 1 mL per well in a 12-well plate. Cells were incubated with siRNA complexes for 48 h prior to HCMV infection.

## Quantitative qPCR analysis

Total RNA was extracted using Direct-zol RNA Miniprep Kit (Zymo Research) following the manufacturer's instructions. cDNA was prepared using the qScript FLEX cDNA Synthesis Kit (Quanta Biosciences) following the manufacturer's instructions. qPCR was performed using SYBR Green PCR master-mix (ABI) on the QuantStudio 12K Flex (ABI). Amplification of NRP2 and ITGB3 was normalized to the host gene ANAX5 (primers detailed in Supplementary Data 5).

For whole cell analysis, total DNA was extracted using QIAamp DNA Blood kit (Qiagen) according to the manufacturer's instructions. For virion DNA extraction, samples were first treated with DNAseI to remove viral DNA not enclosed in intact virions (PerfeCTa, 95150), following the manufacturer's instructions. 0.4 mg/ml proteinase K (Invitrogen, 25530049) was added, and the samples were incubated for 1 h at 60 °C and then 10 min at 95 °C.

Amplification of the viral gene UL44 was normalized to the host gene B2M (primer detailed in Supplementary Data 5).

## RNA in vitro transcription

PDGFRα and THBD were amplified using the primers detailed in Supplementary data 5 and cloned into the DNA template plasmid (Takara IVTpro mRNA Synthesis System) using Gibson reaction and linearized with HindIII-HF (NEB) for following in vitro transcription reaction. IVT RNA was produced using the CleanScribe™ RNA Polymerase (E-0107, TriLink Biotechnologies) according to the manufacturer's instructions. UTP was substituted with N1-methylpseudouridine-5′-triphosphate (N-1081, TriLink) and the RNA was co-transcriptionally capped with CleanCap Reagent AG (N-7113, TriLink). The RNA was treated with DNAseI (ON-109, Hongene), precipitated with LiCl and reconstituted in ddH2O. Primary monocytes were transfected with the produced RNA with jetMESSENGER® mRNA transfection reagent (Polyplus), together with 4 μM Ruxolitinib for 12 h before HCMV infection.

## Western blot

For western blot analysis cells were lysed in RIPA buffer (150 mM NaCl, 1% NP-40, 0.5% sodium deoxycholate, 0.1% SDS, 50 mM Tris-HCl pH 7.4, and 1×EDTA-free protease inhibitor cocktail). Lysates were cleared by centrifugation and supplemented with sample buffer. Proteins were separated by SDS-PAGE electrophoresis, transferred to nitro-cellulose membranes (0.25 mm, ThermoFisherScientific), and detected using an infrared fluorescent antibody detection system (LI-COR) using the antibodies PDGF Receptor α Antibody (#3164, Cell Signaling Technology) and GAPDH (2118S, Cell Signaling Technology) as loading control.

## Mass spectrometry

The samples were lysed and digested with trypsin using the S-trap method. The resulting peptides were analyzed using nanoflow liquid chromatography (nanoAcquity) coupled to high-resolution, high mass accuracy mass spectrometry (TIMS-TOF Pro). Each sample was analyzed on the instrument separately in a random order in DIA mode. The resulting data was analysed using Spectronaut software package using the Direct-DIA workflow.

## Reporting summary

Further information on research design is available in the Nature Portfolio Reporting Summary linked to this article.

## Data availability

All next-generation sequencing data files have been deposited in Gene Expression Omnibus under accession number GSE280650. Mass spectrometry data are available via ProteomeXchange with identifier PXD071694. Source data are provided with this paper.

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

## Acknowledgements

We thank Dr. Orly Laufman, Prof. Yossef Shaul and the members of the Stern-Ginossar lab for the critical reading of the manuscript. We thank Dr. Dor Simkin, Dr. Avner Leshem and Tatiana Smirnova for technical assistance. All optical imaging acquired at the de Picciotto Cancer Cell Observatory in memory of Wolfgang and Ruth Lesser of the Moross Integrated Cancer Center in the Department of Life Science Core Facilities, Weizmann Institute of Science. We would like to thank the Weizmann flow cytometry and microscopy unit for technical assistance. We'd like to thank David Morgenstern from the protein profile unit at the G-INCPM, the Weizmann Institute, for his assistance with proteomics analysis. This study was supported by a European Research Council consolidator grant (CoG-2019-864012) and an Israel Science Foundation grant to N.S.-G. (2507/23).

## Author contributions

Y.K., N.S.-G., and M.S. conceived and designed the project. Y.K., T.A., A.W., and M.S. performed the experiments. T.F. helped analyze the HCMV particle images. K.B prepared the RNA for the transfections experiments. Y.F. offered valuable guidance throughout the research. Y.K., A.N., N.S-G., and M.S. analyzed and interpreted the data. Y.K., N.S.-G., and M.S. wrote the manuscript with input from all the authors.

## Competing interests

The authors declare no competing interests.
