## [Transparent Peer Review File · Nature Communications]

Viral Entry Shapes HCMV Latency Establishment

Corresponding Author: Dr Michal Schwartz

Version 1:

Reviewer comments:

Reviewer #1

(Remarks to the Author)

Review of the revised version of the manuscript "Viral entry shapes HCMV latency establishment" from Kitsberg et al.:

I thank the authors for the addition of new data and the very detailed discussion in their rebuttal. Regarding the PDGFRa part: I appreciate the added data on the PDGFRa expressing THP-1 cell lines. The explanation of the differences in the production of infectious progeny between the two PDGFRa expressing cell lines was well discussed in the rebuttal including citations showing the ability of soluble PDGFRa to inhibit HCMV infection. I do not understand though why this data was not added to the revised version of the manuscript - in my opinion it needs such a detailed explanation of the phenotype together with the cited literature. It is an interesting finding, even though it is not the main focus of the paper. Hence, the data shown in this manuscript regarding the HCMV entry receptor overexpression experiments is sufficient, but needs to be integrated in the main text.

However, the data showing the influence of HCMV entry on latency establishment (and reactivation) is still not convincing me. The figure in the rebuttal showing the data of the PFU assay (last figure in the rebuttal) should be placed in the main text, because it is the only data figure showing that the IE bright population presumably cannot reactivate. Besides showing viral DNA replication and late gene expression in the IE bright population, the most important difference between lytic infection and latency is the potential of latently infected cells to reactivate. In case the IE bright cells would still be able to reactivate significantly, the hypothesis of the authors would not hold true and HCMV entry would only have a minor effect on HCMV latency. Hence, an experiment showing that the IE bright cells do not reactivate is crucial to substantiate the claim that HCMV entry shapes HCMV latency establishment. Without this data, the above claim cannot be made.

Response to reviewer #1 comments:

We would like to thank the referee for his time and thoughtful comments.

In the revised version we have addressed the two issues raised by the reviewer:

Reviewer #1 (Remarks to the Author):

Review of the revised version of the manuscript "Viral entry shapes HCMV latency establishment" from Kitsberg et al.:

I thank the authors for the addition of new data and the very detailed discussion in their rebuttal. Regarding the PDGFRa part: I appreciate the added data on the PDGFRa expressing THP-1 cell lines. The explanation of the differences in the production of infectious progeny between the two PDGFRa expressing cell lines was well discussed in the rebuttal including citations showing the ability of soluble PDGFRa to inhibit HCMV infection. I do not understand though why this data was not added to the revised version of the manuscript - in my opinion it needs such a detailed explanation of the phenotype together with the cited literature. It is an interesting finding, even though it is not the main focus of the paper. Hence, the data shown in this manuscript regarding the HCMV entry receptor overexpression experiments is sufficient, but needs to be integrated in the main text.

We have integrated the data explaining the differences in the production of infectious progeny between the two PDGFRa expressing cell lines into the main text. We also discussed the source of this difference, including the citations showing the ability of soluble PDGFRa to inhibit HCMV infection (Lines 180-188, Figs S4m, 4g and 4h).

However, the data showing the influence of HCMV entry on latency establishment (and reactivation) is still not convincing me. The figure in the rebuttal showing the data of the PFU assay (last figure in the rebuttal) should be placed in the main text, because it is the only data figure showing that the IE bright population presumably cannot reactivate. Besides showing viral DNA replication and late gene expression in the IE bright population, the most important difference between lytic infection and latency is the potential of latently infected cells to reactivate. In case the IE bright cells would still be able to reactivate significantly, the hypothesis of the authors would not hold true and HCMV entry would only have a minor effect on HCMV latency. Hence, an experiment showing that the IE bright cells do not reactivate is crucial to substantiate the claim that HCMV entry shapes HCMV latency establishment. Without this data, the above claim cannot be made.

We have included the data showing that the IE bright cells do not reactivate in the main text (lines 290-291, Fig. S8c).